# Recent Progress of Nanocarrier-Based Therapy for Solid Malignancies

**DOI:** 10.3390/cancers12102783

**Published:** 2020-09-28

**Authors:** Qi-Yao Wei, Yan-Ming Xu, Andy T. Y. Lau

**Affiliations:** Laboratory of Cancer Biology and Epigenetics, Department of Cell Biology and Genetics, Shantou University Medical College, Shantou 515041, Guangdong, China; 19qywei@stu.edu.cn (Q.-Y.W.); amyymxu@stu.edu.cn (Y.-M.X.)

**Keywords:** nanotechnology, drug delivery, solid tumor therapy, drug resistance

## Abstract

**Simple Summary:**

Although conventional anti-cancer drugs have been the footstone in the fight against cancer, yet they are far from optimal due to issues related with indiscriminate destruction of normal cells, multidrug resistance, and toxicity, as a result, a more selective therapy is urgently needed. Nanocarriers have been increasingly used in drug delivery, especially in cancer therapy. Nanocarriers can improve the therapeutic effect of drugs in cancer by enhancing the specificity and prolonging the circulating half-life of drugs. The aim of this review is to offer a detailed description of different cytotoxic drug nanocarriers and their recent progress. It is expected that this review will be of help to those who have been seeking new study directions in this field and also ones who are about to start the research on nanocarrier-based drug delivery.

**Abstract:**

Conventional chemotherapy is still an important option of cancer treatment, but it has poor cell selectivity, severe side effects, and drug resistance. Utilizing nanoparticles (NPs) to improve the therapeutic effect of chemotherapeutic drugs has been highlighted in recent years. Nanotechnology dramatically changed the face of oncology by high loading capacity, less toxicity, targeted delivery of drugs, increased uptake to target sites, and optimized pharmacokinetic patterns of traditional drugs. At present, research is being envisaged in the field of novel nano-pharmaceutical design, such as liposome, polymer NPs, bio-NPs, and inorganic NPs, so as to make chemotherapy effective and long-lasting. Till now, a number of studies have been conducted using a wide range of nanocarriers for the treatment of solid tumors including lung, breast, pancreas, brain, and liver. To provide a reference for the further application of chemodrug-loaded nanoformulations, this review gives an overview of the recent development of nanocarriers, and the updated status of their use in the treatment of several solid tumors.

## 1. Introduction

Chemotherapy, the use of cytotoxic drugs in their free form to kill cancer cells or inhibit cancer cell division, remains the mainstay of treatment for solid tumors, majorly in cases of cancer of the lung, breast, brain, liver, and pancreas. Although antineoplastic drugs have improved patient survival and treatment outcomes, their treatments are not effective enough stems from non-specific toxicity, high metabolism, and unfavorable pharmacokinetics [1,2,3]. Adverse effects will continue to occur in the form of induction of multidrug resistance and low survival profile of patients. Owing to these threats, nanocarriers have been attempted for cancer therapy as quite encouraging systems. Antineoplastics can be entrapped within, physically adsorbed, or form chemically covalent bonds on the surface of the nanocarrier [4].

Nanomedicines have been a highly active field, as exemplified by the steadily-increasing number of articles reporting on nanocarriers for drug delivery (Figure 1). The nanotechnology-based novel drug delivery system (DDS), to some extent, can overcome the above-mentioned drawbacks of chemotherapy drugs mainly based on two mechanisms: Passive and active targeting. Their application as successful drug-carrier for chemotherapeutic drugs in cancer treatment is primarily related to their size, higher surface area to volume ratio, shape, charge, and composition [5]. Certain sized drug-loading particles tend to accumulate in tumor tissue much more than they do in healthy tissues, namely “passive targeting” [6]. Nanosize of NPs enables them to specifically accumulate inside the tumor through enhanced permeability and retention effect (EPR) characteristic of tumors. EPR is based on aberrant pathophysiological characteristics of tumors [7]. The presence of irregular, fenestrated blood vasculature and diminished lymphatic drainage results in the extravasation of NPs from the circulation into the tumor rather than the surrounding healthy tissue and prevented clearance of NPs, leading to their accumulation in the tumor tissue [8]. In other words, the high permeability of the tumor vasculature makes it easier for large molecules and NPs to enter. At the same time, tumor tissues generally lack effective lymphatic drainage, therefore allowing NPs to accumulate there. In order to enhance passive accumulation in the tumor site, the common method is to modify the size or shape of NPs. However, the efficiency of passive targeted delivery is very low, with less than 1% of nanotherapeutic drugs found in tumors [9]. When passive targeting is insufficient, the active targeting can be attained by conjugation of the NPs with targeting ligands (e.g., antibodies or peptides), or through the use of an external stimulus to the desired location [10]. Conjugation of NPs with ligands not only facilitates accumulation in the interstitial space of the tumor, but also receptor-mediated endocytosis leads to the internalization of NPs, providing further opportunities for the release of targeted drugs [11]. Active targeting has more potential than passive targeting strategies, due to it ideally allowing for cell-specific killing of not only primary tumor cells but also of metastatically spread circulating cancer cells [12,13]. Currently, active targeting NPs are supported as a favorable complementary strategy to EPR to further enhance the efficiency of nanodrugs. Stimuli-regulated release NPs systems have been designed to respond to different types of stimuli. The stimuli can be characteristic of the pathological site (internal stimuli), and stimuli-regulated release NPs systems are achieved through the inclusion of components that react to abnormal pH, temperature, and redox conditions, and to the overexpression of certain biological molecules. The NPs systems can also respond to stimuli from outside the body, such as light, ultrasound, and microwaves [14]. These nanomedicines are essentially a multicomponent system, including well-defined nanostructures as delivery vehicles, one or more drugs as a therapeutic agent, and bioactive moieties to extend half-life and promote accumulation in the target site. Thus, the multicomponent system can increase the effective concentration of chemotherapy drugs in tumor cells as well as minimize the risk of resistance.

In this review, we outline the varied architectures of nano-systems and the recent development of nanotechnology in the field of solid tumor treatment. Focusing the direction of this field will help to develop new therapeutic strategies and improve the therapeutic outcome of solid tumors.

## 2. Development of Nano-Sized Delivery Systems for Solid Cancer Therapy

### 2.1. Liposomal Nanocarriers

Among the nanocarriers used to treat cancer, lipid-based nanocarriers have made great progress. There are currently different types of lipid-based formulation, such as liposome systems, solid lipid NPs (SLNs), and nanostructured lipid carriers (NLCs). These lipid-based systems tend to be less toxic than other DDSs, such as polymer NPs, because of their biocompatibility and biodegradability.

Liposomes are artificially-generated spherical drug carrier vehicles composed of a lipid bilayer surrounding a hollow core into which chemotherapeutic drugs can be loaded for delivery to tumors sites [15,16]. The assembly of liposomes is fairly straightforward because the amphiphilic nature of the phospholipid and the thermodynamic properties of the water environment drive the self-assembly into an entropically favorable direction, with a hydrophobic segment enclosed within the NP core [17]. By far, liposomes have been the most successful formulation for clinical application [18]. The bilayer structure of liposomes can be composed of natural phospholipids, cholesterol, etc., making them ideal carriers for different drugs of varying solubility, as hydrophilic molecules can be incorporated within the core (e.g., Doxil^®^, encapsulated doxorubicin (DOX)) while hydrophobic drugs can be housed within the lipid membrane [19]. Thus, liposomes can carry both water-soluble and poorly soluble drugs to a target site. Beyond that, they possess low immunogenicity, low toxicity, and drug protection. To increase circulation time, biocompatible and inert polymers such as polyethylene glycol (PEG) are added to the surface of liposomes, forming a protective layer, which prevents the clearance of the reticuloendothelial system.

SLNs are nanosized lipid-based colloidal carrier systems, combining the advantages of colloidal counterparts (e.g., polymeric NPs, liposomes, and nanoemulsions) [20]. Some areas where SLNs perform better than their counterparts, like economical large-scale production, long-term stability, and controlled drug release [21]. SLNs are the most common method to improve the oral bioavailability of water-soluble drugs. However, the SLNs system is limited by the low drug loading compared with other nanosystems.

NLCs are the second generation of SLNs, which are a mixture of different lipids, i.e., solid lipid matrix with a certain content of a liquid lipid. NLCs have greater drug loading than SLNs, as many drugs have different solubility in solid and liquid lipids. Increasing the liquid-fat quality may increase the solubility of drugs and improve the encapsulation efficiency of lipid carriers [22,23]. NLCs showed a lower risk of gelation and drug leakage, it can also prolong the half-life of drugs, enhance the EPR effect, subsequently improves the therapeutic effect of anti-tumor drugs.

### 2.2. Micelles

Micelles are another type of lipid nanostructure and are defined as a collection of amphiphilic surfactant molecules that spontaneously aggregate in water into a (usually) spherical vesicle [24]. Micelles have hydrophilic heads that form the outer shell, and hydrophobic tails forming the interior that can protect hydrophobic drugs from the external environment [25]. The hydrophobic/hydrophilic structural characteristics of micelles have attracted attention as a DDS, particularly in improving the bioavailability of low water-soluble drugs. However, poor chemical versatility and structural instability are obstacles to their application.

### 2.3. Polymeric Micelles (PMs)

PMs, consist of a hydrophilic shell and a hydrophobic core, which are self-assembled core–shell constructs in selective solvents [26]. The hydrophobic core of PMs can accommodate large amounts of hydrophobic drugs at higher concentrations, while shell not only provides steric stability of nanostructured micelles, but also facilitate their functionalization, allowing the drugs to be delivered to the target site by controlling pH, temperature, and ultrasound, and decorating with peptides or antibodies [27]. Their major advantage over micelles is their great chemical versatility that allows for controlling and modulating both chemical and structural features to improve drug loading capacity as well as drug target specificity [25]. As the carrier of hydrophobic anticancer drugs, PMs are characterized by high drug encapsulation rate, long drug retention time in blood, increased drug permeability, and strong tumor penetration [28].

### 2.4. Polymer-Based Nanocarriers

Polymeric NPs are colloidal systems. They are organic polymer compound assemblies in the form of nanospheres (solid spheres) or nanocapsules (hollow spheres with a void space in the center) [29,30]. They can be composed of natural polymers (e.g., chitosan, hyaluronic acid (HA), cellulose, and corn starch), synthetic polymers (e.g., polyvinyl alcohol (PVA), PEG, and poly (lactic-co-glycolic acid) (PLGA)) [31,32]. The core–shell structure of polymeric nanosystems facilitates the encapsulation of hydrophobic drugs, extends circulation time, and controls drug release. The tight assembly of the outer particle layers in the polymerized nanocapsules allows better retention of the drug and thus enhanced delivery to the tumor site. The physicochemical characteristics of the polymer (e.g., surface charges, size, shape, flexibility, and length of the main carbon chain) can also be easily engineered to achieve high biodegradability, high content of drug loads, and target tumor locations [33,34]. However, polymer NPs also have some disadvantages, such as residual organic solvents in the preparation process, difficulty in large-scale industrial production, and polymer cytotoxicity.

### 2.5. Dendrimers

Dendrimers, normally 1–10 nm in diameter, are chemically synthesized highly branched polymers with a highly symmetric spherical shape [35]. Usually, they are produced from natural or synthetic ingredients, including sugars, nucleotides, and amino acids. Drugs could be entrapped in the dendrimer core by hydrogen bonds, electrostatic or hydrophobic interactions. Besides, hydrophobic or hydrophilic anticancer drugs can also be covalently attached to the surface of dendrimers. When the drug is attached to many peripheral groups of a dendrimer, it will result in the enhanced solubilization of the drugs, the amplified effective concentration of the drugs at the target site, and the controlled release of the drugs depending on the linkers used [36]. The structure of low-generation dendrimers is usually flexible and open, while high-generation dendrimers are dense and globular [37]. Dendritic polymers are easy to be functionalized and have unique advantages, including high stability, water-solubility, decreased immunogenicity, and antigenicity, which make them an attractive drug delivery carrier. Despite these promising characteristics, dendrimers share a limiting feature with polymer therapeutics: The multistep synthesis that increases production costs [38].

### 2.6. Niosomes

Niosomes, non-lecithin carriers, have a similar structure to liposomes but they are more stable [39]. They are vesicle systems synthesized by nonionic surfactants with some advantages, such as biodegradability, biocompatibility, and encapsulation of both lipophilic and hydrophilic drugs [40]. They were designed to overcome the limitations of liposomes, especially those associated with phospholipid oxidation. At the same time, niosomes have lower manufacturing costs and longer shelf life, and their bilayer fluidity and microviscosity could be easily regulated. pH (Low) insertion peptide (pHLIP)-coated niosomes were successfully developed by Pereira et al. [41]. Compared with pHLIP-coated liposomes, pHLIP-coated niosomes are smaller and more stable, with pH-dependent cell uptake and excellent tumor targeting. Therefore, niosomes loaded with lipophilic and/or hydrophilic drugs can effectively enter into cells in a pH-dependent manner. The versatility of niosomes enhanced drug oral absorption. On the other hand, the encapsulation ability of niosome is relatively low, thus different surfactant combinations are needed to encapsulate various hydrophobic molecules in its bilayer membrane, so as to maintain the overall stability of the nanovesicles.

### 2.7. Nanoemulsions

Nanoemulsions are made of oil, water, emulsifier, and co-emulsifier in proper proportion, and the particle size is 10–100 nm diameter [42]. Nanoemulsions have been widely studied as drug carrier of lipophilic chemotherapeutics, due to its biodegradability, easy preparation, and controllable drug release [43,44]. Besides, nanoemulsions not only can avoid drug inactivation in the gastrointestinal tract but also increase the solubility of the drugs so that the drugs can be well dispersed and absorbed, so as to improve the bioavailability of drugs. Nanoemulsions also has good biocompatibility due to the incorporation of a generally recognized as the safe grade of excipients, in which the entrapment efficiency of the hydrophobic components is high, showing physicochemical stability and improved bioavailability with superior efficacy and safety.

### 2.8. Nanocrystals

Nanocrystals are referred to as pure solid particles with the character of the crystal [45]. Some poorly soluble drugs have been salvaged through formulating nanocrystal. Nanocrystals possess unique traits such as the increased surface area to volume ratio, steady dissolution rates, enhanced structural stability, and high drug-loading efficiency, due to nanocrystals consisting of entirely of the drug or payload, thereby eliminating the ancillary role of a carrier and resulting in satisfactory therapeutic concentrations at low dose [46]. Nanocrystals were originally used to enhance the oral bioavailability of low-soluble drugs. To date, although researches on the drug nanocrystal in cancer treatment are still at the preclinical animal level, nanocrystal formulation has gained wide concern for intravenous delivery of anticancer drugs [45,47]. Due to rapid ingestion by macrophages, intravenous nanocrystals can be passively transported to mononuclear phagocytic system cells rich organs such as the liver, spleen, and lungs [48]. The particle size, morphology, and surface modification of nanocrystals have a great influence on their distribution in vivo. The size, stability, solubility, and bioavailability of nanocrystals are often affected by the pH of the dispersed medium, impurities formed during manufacturing, as well as the crystallinity [46].

### 2.9. Bio-NPs

Due to good biocompatibility, better stability, and biodegradability, nanostructures with biological materials have become a powerful platform for drug delivery [49]. However, the synthesis strategies of bio-NPs could sometimes be complex, resulting in increased cost and manufacturing time. Therefore, more researches are needed to explore the manufacture of these bio-NPs from lab-scale to commercial industrial scale. Viral NPs obtained from viruses and bacteriophages have attracted considerable interest for drug delivery due to their flexibility in sizes and shapes, biocompatibility, and easy surface modification [50].

Nanocarriers synthesized by synthetic polymers at times cause high residue of solvents and surfactants, which may limit their efficacy and induce toxicology issues. In this regard, protein-encapsulated nanocarriers, prepared from animal-based protein (e.g., albumin, collagen, and gelatin) or plant-based protein (e.g., ferritin), have been recently investigated for the drug delivery [51]. Protein-based nanocarriers have several advantages in drug delivery, such as good biodegradability, easy surface modification ability, no immunological responses, and low toxicity. Specifically, the application of albumin-based NPs in the biomedical field has been studied since 1972 [52]. The first protein-based NP approved by the U.S. Food and Drug Administration (FDA) is Abraxane^®^, albumin-bound paclitaxel (PTX) NP formulation, used in combination with chemotherapy drugs to treat a variety of cancers. However, protein drugs usually have pharmacokinetic and pharmacological defects, such as instability, short half-life, and sometimes low water solubility. The protein conjugated with a biologically inert polymer is a good manner to improve its chemical stability while maintaining the biological function. Apoferritin (APO), the hollow protein nanocage, undergoes a process of assembly and disassembly with the change in pH after the iron core is removed and is extensively used to synthesize various NPs for cancer treatment [53].

### 2.10. Exosomes

Exosomes, nanometer-sized extracellular vesicles, are secreted by all types of cells and can also be found in various body fluids [54]. Water-soluble drugs can be stored by the hydrophobic core of exosomes, which are therefore emerging as a promising natural vehicle for drug delivery. Also, exosomes exhibit a natural targeting capacity, high stability, high capacity to pass through various biological barriers, and are less immunogenic than artificial drug carriers, probably due to their small size, and they are isolated from the patient’s cells [55]. Exosomes provide a more stable environment for therapeutic drugs, and their components, cargo, and targeting ability can be further enhanced by conditioning parental cells or adding functional drugs that improve their natural potential, thus giving them additional functions [56]. Even though there are many advantages of exosomes, there are several important obstacles to overcome before clinical application. Natural exosomes are complex structures, making the identification of them difficult [57]. In addition, exosomes may contain inherently numerous bioactive molecules. If the alteration of the exosomes’ cargo is required, it is important to use specific approaches to load the desired additional drugs without disturbing the exosomes, as damaged exosomes with no exosomal signals may lead to undesirable immune responses [58]. Understanding the function and structure of exosomes will contribute to the clinical setting of exosome therapy [59].

### 2.11. Inorganic NPs

Inorganic NPs, which refer to nanocarriers synthesized by metal and semi-metal materials, have gained increasing interest in the recent past. Due to easily scalable synthesis, simplified modification of targeting molecules, high stability, controlled release of the drug, and capability of facilitating targeted drug delivery with imaging possibilities, inorganic nanomedicines have been studied to find a nanocarrier for delivering chemotherapeutic drugs. Among various inorganic NPs, metal NPs (e.g., gold, silver, ZnO), silica NPs (SiNPs), selenium NPs (SeNPs), magnetic NPs (MNPs), quantum dots (QDs), carbon nanotubes (CNTs), and nanodiamonds (NDs), were widely investigated in cancer treatment.

Many NPs, such as QDs, metal NPs, ceramic NPs, oxide NPs, are difficult to produce in aqueous media or require multiple processes and difficult to control reactions, so consistency in products is not always guaranteed. Au NPs, an attractive challenger in cancer therapy, possess unique properties including ease of production and functionalization, and high biocompatibility. Silver is usually a non-toxic, hypoallergenic metal with the ability to protect cells and promote healing, and has recently attracted attention as a nanocarrier. In cancer cells, AgNPs induce the ROS generation, which results in inflammatory responses and subsequently mitochondrial destruction. Further apoptogenic factors are excluded leading to the death of cancer cells [60,61]. The toxicity of NPs can be overcome by synthesizing NPs by the biological method and coating the surface of NPs with the degradable non-toxic polymer.

SiNPs have a significant advantage in oral administration. In the complex gastrointestinal environment, SiNPs can both protect hydrophobic drugs from the intestinal milieu and be resistant to low pH. In addition to non-porous SiNPs, porous SiNPs characterized by adjustable pore size, ordered pore structure, high pore volume, low-pH tolerance, high thermal stability, and large surface area for functionalization, which has attracted extensive attention. Mesoporous SiNPs (MSNs) are efficient systematic DDS due to the high amount of drug loading capacity, a strong affinity for head groups of different phospholipids, high cellular penetrations, protection of the water-insoluble drug at therapeutic levels, and sustained release at the targeted delivery site. Studies have proven that MSNs can be endocytosed by cancer cells, non-cancer cells, or macrophages.

SeNPs have attracted wide attention as potential chemotherapeutic carriers because selenium is a trace element and plays an important role in cancer prevention [62,63,64]. In addition, selenium facilitates drug toxicity reduction, the regulation of the function of the thyroid gland and the immune system, thus plays a strong part in fighting disease. However, the absence of active targeting capability is still a problem to be solved.

MNPs are a class of nanoscale carriers containing iron oxide. The discovery of MNPs has attracted considerable research interest, primarily because of their ability to perform multiple functions simultaneously, such as colloidal carriers that can target drugs to tumor sites with real-time monitoring. The simplest form of drug delivery of MNPs consists of an inorganic material core and a decorated surface coating to enhance stability and biocompatibility under the physiological environment.

QDs are a kind of tiny inorganic semiconductor nanocrystals with a diameter of 1–10 nm. Due to its unique surface chemistry available for modification and multi-wavelength luminescence characteristics of high luminous stability, they have attracted much attention in tumor research and become an ideal material for targeted drug delivery [65]. The general structure of QDs is composed of a semiconductor core, coated with a shell to change its physical and chemical properties and improve solubility [66]. However, the major disadvantages of QDs are its toxicity and excretion pathway. Researches have shown that their toxicity depends largely on their core–shell materials, bioconjugation, and surface functionalization.

CNTs are carbon allotropes that are insoluble in water and other organic solvents, and their toxicity in biological liquids is a key limiting factor. However, they can be transformed into water-soluble nanocarriers by chemical modification to improve their biocompatibility and reduce toxicity. The material has unique physicochemical properties, as the hollow monolithic structure that can accommodate high payload and the ability to add any functional groups, making them a suitable and effective delivery system for chemotherapeutic agents [67].

NDs include unique electrostatic properties, a chemically inert core, scalable processing and synthesis parameters, and an adjustable surface resulting in a simple mechanism of drug deposition [68]. NDs as drug carriers have attracted much attention, due to the facile nature of functionalizing their surfaces with chemotherapeutic drugs, especially anthracyclines. The drugs can be both covalently or noncovalently attached to NDs [69].

Certain other inorganic NPs, such as bismuth, barium, calcium, magnesium, copper, nickel, and titanium dioxide, are also used in the treatment of cancer [70]. The main risk of these inorganic NPs is toxicity, which is closely related to the size control, shape regulation, surface modification, concentration, and time of exposure. The precise control of these physicochemical parameters can promote their meaningful application in cancer therapy. Nonetheless, clinical translation of these nanomaterials is still not extensive compared to other counterparts, due to a lack of understanding of their long-term toxicity, pharmacokinetics, pharmacodynamics, and degradability.

### 2.12. Hybrid Nanomedicines

Hybrid nanomedicines are a mixture of inorganic and organic ingredients that enables the desired hybrid nanomaterial formulation and allows the system to change to achieve the desired results. Current researches focus on the mixing of polymer and liposome systems. The resultant structures, known as polymer–lipid hybrid NPs (PLNs), are characterized by particle sizes less than 100 nm in diameter. The core of PLNs is a polymer–drug complex and the increasing number of bio-polymers offers a number of opportunities for drug conjugation. The outer shell of PLNs is composed of lipid, which contributes to their high biological compatibility [71]. Thus, PLNs combine the biomimetic properties of liposomes with the sturdiness of biodegradable polymers, which display high structural integrity, controllable drug release, and the possibility of binding to targeted delivery factors.

Lipid nanocapsules (LNCs) consist of a liquid oily core and a shell of surfactants, as a hybrid between polymeric nanocapsules and liposomes [72]. The LNCs can contain comparatively high concentrations of liposoluble drugs in their liquid oily core. Unlike earlier nanocarriers, LNCs are obtained with pharmaceutically acceptable excipients based on a solvent-free soft-energy procedure. Moreover, LNCs have a high drug-loading capacity and physical stability, and a sustained drug release pattern [73]. In a word, LNCs are promising nanocarriers for drug delivery.

Metal–organic frameworks (MOFs, also known as porous coordination polymers), a unique class of hybrid porous materials, which have attracted significant research interest in biomedical application. MOFs are built from metal ions and organic linkers [74,75]. Due to their chemical versatility, enormous porosity, high drug loading, and tunable degradability, nanoscale MOFs have been adopted as promising carriers for therapeutic drugs [76,77].

Schematic representation of the commonly used as nanocarrier types were showed in Figure 2. Summary of types of NPs and their advantages and disadvantages were included in Table 1.

## 3. Encapsulation of Anti-Cancer Drugs with NPs

Over the past three decades, anthracyclines and taxanes are two mainstay chemotherapeutics against cancer [78]. Anthracyclines which include DOX, epirubicin, and daunorubicin, among others, are potent DNA intercalating agents and result in DNA damage triggering apoptosis in cells by DNA interaction [79]. They are effective against cancer cells but will kill both normal and cancer cells. Among the anthracyclines, DOX is a primary therapeutic agent in combination regimens for the treatment of lymphomas, breast cancer, osteosarcoma, and other solid tumors [2,80]. Taxanes prevent the growth of cancer cells by destroying microtubules, so cancer cells cannot grow and divide [81]. Among the taxanes, PTX is a tetracyclic diterpene compound and used in metastatic pancreatic adenocarcinoma, breast cancer, non-small cell lung cancer (NSCLC), and ovarian cancer, while docetaxel (DTX) was licensed for the treatment of early stage and metastatic breast cancer [82]. However, the poor water solubility of taxanes limits their clinical success. DTX, a semisynthetic taxane, binds, and stabilizes microtubules by inducing G2/M arrest, thus interrupts cell division and inhibits tumor growth [83,84]. However, the clinical application of DTX has been hampered due to a number of harmful reactions, including anaphylaxis, neurotoxicity, neutropenia, peripheral neuropathy, and musculoskeletal toxicity [85]. Gemcitabine (GEM) is a fluorine-substituted deoxycytidine analogue with broad spectrum antitumor activity and its mechanism of action is based on the irreversible inhibition of DNA synthesis. However, GEM has a short half-life, and rapid body clearance, thus usually administered in higher and repeated doses, leading to many side effects [86]. It has also been demonstrated to rapidly induce drug resistance in cancer cells via many different but unclear mechanisms [87]. Carmustine (CMS) is a cell-specific nitrosourea alkylating agent, which can inhibit DNA repair and RNA synthesis in glioma cells. Due to its high lipophilicity, CMS can penetrate the blood–brain barrier (BBB) [88]. However, there are several limitations of using CMS: Short half-life, lack of selectivity to tumor cells, and short retention in the brain [89].

Drugs employed in the treatment of cancer can be loaded either by entrapment within, adsorption on, or by covalently binding to the nanoparticle [4]. As shown in Figure 3, drugs can be attached to the outer shell or the corona of NPs by covalent bonding or physical adsorption as for example by electrostatic interactions between the nanoparticle and drug, which usually exhibit low stability and become pH liable [90,91]. The non-covalent binding is simple, versatile in application, and the structure and biological activity of the drugs is exposed to minimal change. In addition, the drug releases easily and quickly in response to environmental stimuli. When the drug links to the nanocarriers by the covalent bond, leading to the enhanced solubilization of the drugs, the amplified effective concentration of the drugs is significantly amplified at the target site, and the controlled release of the drugs depending on the linkers used [36,92,93]. However, compared with free drugs or drug carrier complexes, covalent drug–carrier complexes have lower anticancer activities of chemotherapeutics. Drugs also could be entrapped into the core of NPs by hydrogen bonds, electrostatic, or hydrophobic interactions. Hydrophobic drugs possibly produce a hydrophobic interaction between the drug and the core of the particle, increasing its solubility [94]. The encapsulation of anticancer drugs in NPs is probably best utilized in local treatments (intratumoral injections), because, although it solubilizes the hydrophobic drugs and leaves the drug unaltered, results in toxicity and a rapid drug release may occur before reaching the target site [95].

## 4. Application and Clinical Trials of Nanocarrier-Based Therapy in Solid Tumors

Targeting NPs, based on the unique physical and chemical properties of the tumor microenvironment, have been widely studied for the treatment of tumors. A brief summary of the application of nano-based DDS in selected cancers is given in Table 2. Typically, the nanomedicines are designed to attack a certain molecular agent or pathway involved in the development of cancer. The selection criteria for the NPs showed in Figure 4.

### 4.1. Lung Cancer

Lung cancer is an aggressive type of cancer with the highest morbidity and mortality. Lung cancer is divided into small cell lung cancer (SCLC) and NSCLC, of which NSCLC is the most common. Despite the emergence of a variety of treatment methods, there is no denying that chemotherapy is still the dominant clinical treatment for lung cancer, and immunotherapy has emerged as a potent additional treatment against lung cancer [142,143]. However, current treatments for lung cancer have met with limited success, due to the poor targeting of chemotherapeutic drugs, therapeutic resistance, heterogeneity, and the high metastasis of cancer cells [144,145]. Thus, one alternative strategy, active targeting via NPs, has shown promise in the treatment of lung cancer. Previous studies have shown that EGFR up-regulation is often associated with NSCLC, thus EGFR can be used as a targeted molecule to target the delivery of chemotherapy agents [146]. CD44, Sigma, DR 4/5, transferrin, and glucose receptor were also used as a targeted molecule, which can deliver anticancer drugs to tumor cells without harming normal cells [147,148]. Besides, CD133 and CD44 were also specific markers for lung cancer-initiating cells [149]. Attempts have been made to deliver chemotherapeutic drugs by exosomes, incorporation of PTX into exosomes (exoPTX) increased cytotoxicity more than 50 times in drug resistant MDCK_MDR1_ (Pgp+) cells. Animal studies have shown that a nearly complete co-localization of airway-delivered exosomes with cancer cells in an LLC mouse model, and a potent anticancer effect in this mouse model [96]. A study reported improved delivery of conventional chemotherapeutics by using QDs that had very low toxicity as nanocarrier [97]. Likewise, a mixed micelle system based on mPEG-SS-PTX and mPEG-SS-DOX conjugate has a significant cytotoxicity to A549 cells [98]. In another study, HA-modified SeNPs loaded with PTX (HA-Se@PTX) displayed significant cellular uptake and controlled release of PTX in vitro [100]. In an attempt to achieve dual-target, Huang and coworkers developed CD133 and CD44 aptamer-conjugated nanomicelles loaded with gefitinib (CD133/CD44-NM-Gef) that were capable of simultaneous targeting to CD44+ and CD133+ lung cancer-initiating cells. CD133/CD44-NM-Gef displayed greater therapeutic efficacy against lung cancer-initiating cells which is crucial for improving therapeutic effects [102]. Recently, a smart PLGA system (PLGA-SS-PEG) loaded with HHT and targeted EGFR has been evaluated in vitro and in vivo, which showed this nanotherapeutic strategy to be safe with better targeting effect. As a targeted nanomedicine, PLGA-SS-PEG is a polymeric drug carrier and targets cancer cells by using an EGFR aptamer [103]. Immunotherapy has become an effective additional therapeutic strategy against lung cancer. Pulmonary surfactant (PS), related to local inflammation and immune response, is abundant in the lung making it different from other organs [150,151]. Pluronic P105 can interact with PS through van der Waals forces and hydrogen bonding [152]. Amphiphilic polymers polyethyleneglycol-polylactic acid (PEG-PLA) and Pluronic P105 were employed as nanocarriers to encapsulate PTX to form into PEG-PLA/P105/PTX micelles. Preclinical studies showed that PEG-PLA /P105/PTX micelles respond to the biological functions of Ax that promotes the secretion of PS and inhibit autophagy, thus modulating the tumor microenvironment to improve drug transportation and cell-killing sensitivity of the micelles [104].

Recently, combinatorial treatment approaches have shown great potential to enhance the therapeutic efficacy, as they better address tumor heterogeneity. A new study was carried out using cell membrane protein-based biomimetic NPs encapsulated DOX and icotinib to EGFR-mutant NSCLC, which achieved high drug accumulation in tumors and enhanced cytotoxicity of the chemotherapeutic drugs. The animal experiment result showed the H1975 cell membrane-coated NPs resulted in 87.56% tumor inhibition, with the tumor weight 8.75-fold less compared to that of the PBS control group [101]. In another study, DQA modified micelles loaded with PTX and ligustrazine was synthesized to inhibit tumor metastasis. Results showed that the inhibitory effects of DQA modified PTX plus ligustrazine micelles on A549 cell invasion were better than PTX plus ligustrazine micelles. Moreover, DQA modified PTX plus ligustrazine micelles showed the strongest adhesion inhibition and the down-regulation effect on metastasis-related proteins, which proved that the micelles could effectively inhibit the process of tumor metastasis. Improved cellular uptake and drug accumulation in tumor tissue were also observed in vivo [99]. Recently, a study was conducted using PTX and GEM-conjugated NLCs on the NSCLC cell lines to determine its efficacy against cancer. In vitro release studies showed that a sequential release of drugs, first PTX (redox-triggered), then GEM (pH-triggered). In addition, this study indicated this nanomedicine achieved a synergistic antitumor effect by targeted intracellularly sequential drug release in vivo [105].

### 4.2. Breast Cancer (BC)

BC is a highly malignant tumor and the most common cause of death in women [153]. The high death rate of BC suggests that current drug treatments are far from optimal [154]. Generally, breast cancer therapeutic agents are administered intravenously or orally and the drug must pass through many barriers to reach the target tumor [155]. Various chemotherapeutic drugs are used to treat breast tumors; however, some patients do not respond to these products originally designed for general anticancer purposes [156,157,158]. Reasons for the failure in breast cancer treatment are high heterogeneity, cancer cells use drug delivery pumps to throw away the drugs which are inside the cell, metastasis that is not affected by drugs, and stem cells develop resistance to chemotherapy [159,160].

This disease is defined by the expression of estrogen, progesterone, and HER-2 [161]. Breast cancer cell receptors play a strong part in the treatment of this disease, as it forms the basis of a targeted strategy for treatment. Besides HER-2, some other molecular targets have been used in the active targeting of BC. An important receptor is the EGFR overexpressed in up to half of the BC cases and has a high density on the cell surface [162]. FR is also common targets for drug delivery, as FR is expressed in 50–86% of metastatic TNBC patients who generally have poorer prognosis [163]. Transferrin receptors, Fn14, estrogen receptors (ERs), and CXC chemokine receptor type 4 (CXCR-4) are also used in a targeted strategy for treatment [164,165,166,167]. Han et al. fabricated PEG long-circulating liposomes loaded with PTX for targeting ERs in breast cancer. The liposome formulation can effectively target, quickly, and specifically identify the tumor site, and prolong drug action time [116]. Targets of breast cancer stem cells (BCSCs) include CD44 and CD133 receptor [168,169]. Besides anti-CD44 monoclonal antibodies, several nano-delivery systems have been developed to target CD44 receptors using different targeting moieties, such as HA [168]. In recent years, with an in-depth understanding of the molecular biology of BC, a number of promising nano-therapeutic strategies have been developed [170]. In a study, pegylated PTX nanocrystals (PEG-PTX-NCs) were prepared and the antitumor efficacy of PEG-PTX-NCs was investigated. The animal experiment results showed that PEG-PTX-NCs significantly enhanced the antitumor effect in treating in situ tumor or metastatic tumor bearing mice after intravenous administration [106]. In another study, the efficacy of cisplatin loaded into ultra-short single-walled CNTs capsules was investigated in vivo. Results indicated that this nanodrug showed a prolonged circulation time compared to free cisplatin which EPR effects resulting in significantly more cisplatin accumulation in tumors [107]. Recent research has reported the use of plant viral NPs for delivering DOX. The finding suggested that DOX-loaded viral NPs were effective in MDA-MB-231 cells although at lower efficacy than free DOX [109]. Additionally, a study investigating the efficiency of DOX loaded SiNPs (SiNPs/DOX). Findings showed that the SiNPs/DOX improved the efficiency of cellular drug delivery, exhibited high cytotoxicity, and successfully inhibited the tumor growth [110]. In another study, the researchers reported that a cross-linked multifunctional polymeric NPs loaded with DTX (DTX-CMHN) showed significantly better inhibition of primary 4T1-Luc tumor growth and lung metastasis with little body weight loss compared to the free DTX group in vivo [112]. Difficulty in eradicating cancer stem cells (CSCs) is another main cause of failure in the BC treatment. Fortunately, multiple studies have shown that dual-drugs (PTX, DOX, 5-fluorouracil (5-Fu) and dexamethasone) simultaneously loaded NPs are effective against BC stem cells. Findings suggested that the dual-DDS, especially carcinogenic drugs in combination with plant and other natural source compounds, has better effects with reduced toxicity in the treatment of BC [160]. In addition, salinomycin-loaded PLNs anti-HER2 NPs (Sali-NP-HER2) were developed to target both HER-2-positive BCSCs and cancer cells. The study indicated Sali-NP-HER2 efficiently targeted to HER2-positive BCSCs and cancer cells, resulting in enhanced efficiency compared with non-targeted NPs or salinomycin [108].

The emergence of resistance to chemotherapy in BC is one of the major obstacles to achieve the success of BC treatment. To address drug-resistant BC, an arginine-glycine-aspartic (RGD) tripeptide coated, pH-sensitive SLNs (RGD-DOX-SLNs) were employed to load DOX. RGD-DOX-SLNs showed a higher area under the plasma concentration-time curve and peak concentration compared to DOX solution with no obvious toxicity on cell [111]. Likewise, a liposomal co-delivery system co-loaded with DOX and poria cocos extract was prepared. Results indicated that DOX and poria cocos extract synergistically reversed multi-drug resistance (MDR) during tumor treatment with decreased cardiac toxicity [113]. In another study, a new polymeric micelle composed of phenylboric acid (PBA)-modified F127 (active-targeting group) and DOX-grafted P123 (prodrugs group) (FBP-CAD) was designed for enhancing tumor MDR reversal and chemotherapy efficiency in BC. Results revealed that FBP-CAD micelles possessed stronger cell-killing capacity in vitro, and specifically accumulated at the tumor site with decreased cardiotoxicity in vivo [118].

It is well known that EPR can enhance targeted drug delivery especially for solid tumors (e.g., BC). Without any specific receptor target, the nanocarrier (<100 nm in diameter) can penetrate the cells at the cancerous site through endocytosis, increasing the availability of drugs acting on intracellular organelles. Through EPR, nanocarriers have enhanced the anticancer effects of PTX and DOX, since nanocarriers can passively enter cancer cells and act on intracellular targets [165]. In the latest study, DOX is loaded into the zeolitic imidazolate framework, leading to effective drug accumulation in tumors due to the EPR effect and precise release of the drug in the tumor site by its pH sensitive instability with no side effects to the normal tissue [114].

In addition, the efficacy can also be enhanced by binding active target ligands on the particle surface, increasing tumor selection, and tumor cell-specific uptake drug. In one study, a PTX loaded nanoscale polymer was designed to specifically target the cell surface receptor Fn14. Targeting to Fn14 enhanced the inhibition of breast tumors and significantly minimized the nonspecific binding to blood serum proteins and tumor tissue components [115]. Another study reported an RNA four-way junction NPs with ultra-thermodynamic stability covalently loaded with high-density PTX (RNA-PTX) for targeted cancer therapy. Results showed that RNA-PTX dramatically strongly accumulated in tumors and inhibited tumor growth with negligible toxicity in vivo [91]. Zafar and its associates developed pegylated LNCs for co-encapsulation of DTX and thymoquinone (THQ). The in vivo tumor growth inhibition study showed that the average tumor volume was lowest for the pegylated nanocapsules treated group compared to the control group and free DTX treated group, which confirmed the anti-cancer superiority of the dual drug-loaded nanocapsules. In addition, the formulation showed a remarkable reduction in toxicities associated with the liver, kidney, and oxidative stress [117].

### 4.3. Pancreatic Cancer (PC)

PC is one of the most lethal solid malignancies, mainly due to its dense fibrotic stroma, high metastasis, and highly immunosuppressive tumor microenvironment limiting the therapeutic efficacy of available chemotherapeutics and has low cure rate [171]. The desmoplasia of the stroma of PC takes up most of the tumor mass (80% or more), leading to abnormal vascularization, high intratumoral pressure, and poor drug diffusion [172]. Therapeutic agents administered intravenously must first extravasate and pass through a thick, fibrous tissue to locate a tumor target, thus therapeutic efficacy is limited. High frequency of genomic changes seen in PC results in significant genomic instability and loss of their suppressor functions, reducing response to treatment [173]. There is an intricate network of signaling and genetic alternations and cross talk between cells and microenvironment that make it harder to treat.

Standard chemotherapeutic agents have been used to treat PC include GEM, 5-Fu, leucovorin, irinotecan, oxaliplatin, cisplatin, and capecitabine. Among them, GEM is the frontline drug for treating PC, but it is only effective in 23.8% of these cases [174]. Other drugs are also used for treating PC, including Abraxane^®^, Onyvide^®^, and so on. As the vast majority of patients suffering from PC prefers to chemotherapy, novel and effective chemotherapy drugs are urgently needed.

In order to ensure a high-affinity interaction with cancer cells, antibodies, or receptor agonists were coated to the surface of the nanocarrier. The expression of several surface receptors, such as targeting transferrin receptors, FR, EGFR, Lf receptors, vascular endothelial growth factor (VEGF), and HA receptors, has been associated with the progression of PC [175,176,177,178]. In another study, the tumor homing peptide tLyp1 functionalized HA nanocapsules loaded with DTX, which dual-targeted tumors and the lymphatics. In vitro study showed these nanocapsules could interact with the NRP1 receptors over-expressed in cancer cells. The results showed a dramatic accumulation of DTX in the tumor and a reduction of the tumor [119]. Accordingly, a poly-L-lysine coated PTX loaded PLGA NPs formulation showed the inhibition of tumor growth and anti-metastasis against PC [122]. To eliminate pancreatic CSCs for preventing metastasis, Lf and HA double-coated lignosulfonate (LS) based nanosystem was developed to target Lf receptors on cancer cells and CD44 receptor overexpressed in tumor and pancreatic CSCs. After loading QC, the nanomedicine showed the great inhibition of migration and invasion of PANC-1 cells in vitro and tumor volume reduction in vivo [125].

In addition, the therapeutic effect of PC will be improved from combined treatment. In a study related to the use of liposomes in drug delivery, Lin et al. investigated the effect of GE-11 peptide conjugated liposome loaded with GEM and HIF1α-siRNA (GE-GML/siRNA) on PCCs. According to the results of the experiment, the designed GE-GML/siRNA increased the intracellular concentrations in the cancer cells and showed a significant reduction in the tumor burden. Furthermore, a synergistic combination of GEM and HIF1α-siRNA significantly inhibited cancer cell proliferation [120]. Inspired by this result, ultra-small pegylated NDs loaded with irinotecan and curcumin (ND-IRT + CUR) were studied, and it exhibited superior anti-tumor effects in vivo in two different mouse models of aggressive PC [123].

Regulating the surface hydrophilicity of nanocarriers can prevent them from being eliminated by macrophages, thus prolonging the drug circulation time in the body. The most common modifications are PEG, poloxamers, polysaccharides, and other hydrophilic polymers. In the latter study, PEG-functionalized NDs loaded with DOX (ND-PEG-DOX) displayed excellent biocompatibility and prolonged drug retention in vivo [121]. According to Elechalawar et al., gold NPs were used for targeted delivery of GEM to PCCs with the monoclonal antibody cetuximab (C225) as the targeting ligand for EGFR and PEG as a stealth molecule. As a result, enhanced selectivity towards PCCs and pancreatic stellate cells (PSCs) was observed in vitro [126].

The highly immunosuppressive tumor microenvironment and dense fibrotic stroma barrier are the two major obstacles for achieving great therapeutic efficacy. In order to solve this problem, a study has reported that a redox-responsive GEM-conjugated polymer (PGEM) co-loaded with PTX and NLG919 showed deeper penetration in PC tumor tissues. In vivo studies suggested that incorporation of NLG919 into the nanomedicine reversed indoleamine 2, 3-dioxygenase (IDO)-mediated immunosuppression, and thereby enhanced the therapeutic effect [124]. In another study, an innovative combination of metformin (MET) with pHLIP co-modified Fe_3_O_4_ NPs loaded with GEM has been investigated. MET was firstly administrated to destroy the dense fibrotic stroma barrier, which improved the delivery efficiency of GEM in vivo. Findings suggested that targeted delivery and effective accumulation of the nanomedicine both in vitro and in vivo. Furthermore, the configuration change of pHLIP is controlled by the acidic environment of the tumor, which is beneficial to deliver GEM into the tumor sites [127].

### 4.4. Glioblastoma (GBM)

GBM is the most angiogenic and highly lethal brain tumor, treatment of gliomas is still difficult to achieve satisfying outcomes, as the inability to achieve therapeutic agent concentrations at the tumor tissue. In addition to the genetic and signaling heterogeneity, the BBB and blood–brain tumor barrier (BBTB), known as the major obstacles, make therapy greatly inefficient.

In GBM, the nanomedicines have to be able to cross or bypass the BBB, and at the same time must not cause an immune response [179]. Among these different nanocarriers, polymers have met the strict requirements required for biological applications. For instance, Meng et al. found that a borneol-modified nanomicelle loaded with DOX exhibited superior transport efficiency of DOX across the BBB and accumulation in the brain tissues [131]. Likewise, a study suggested the usage of positively charged chitosan-coated pegylated NPs for co-delivery of R-flurbiprofen and PTX to glioma tissue. It was shown that this PLGA NPs are able to carry their payloads to glioma tissue, enhance the anti-tumor activity, and reduce inflammation in the peri-tumoral area [134].

A more recent approach for increasing anti-glioma efficacy and safety involves the use of a dual-targeting glioma DDS to increase the accumulation at the glioma site after effectively crossing the BBB and BBTB. An example of a receptor-ligand targeting subset of active DDS is the TfR1. A dual-targeted delivery system (GKRK-APO) made of APO and GKRK peptide ligand, increases the specific targeting to brain endothelial cells and glioma cells and displayed higher glioma localization. Based on the results of the experiments, it has been demonstrated that GKRK-APO loaded with VCR efficiently overcame multiple barriers (e.g., BBB and BBTB) and showed an effective anti-glioma effect in vitro and in vivo [128]. Another study relates to the Pep-1&borneol-bifunctionalized CMS-loaded micelles (Pep1/Bor/CMS-M) was able to target interleukin-13 receptor overexpressed glioma and penetrate BBB. In addition, it showed that Pep1/Bor/CMS-M enhanced CMS accumulation in glioma, suppressed the tumor growth, and improved the survival period with low systemic toxicity [129]. It is known that CSCs play an important role in the development and metastasis of tumors and as a barrier against the anticancer effects of chemotherapeutic drugs. Thus, a DDS could significantly enhance anti-cancer efficacy by precisely targeting to both cancer cells and CSCs after effectively crossing the BBB. Recently, it has been reported that SKN and DTX co-loaded nanoemulsion bifunctionalized with AS1411 aptamer, and HA penetrated the BBB, inhibited tumor growth, and prolonged the survival period in vivo. The nanoemulsion was accurately delivered to the glioma region through the high affinity between HA and CD44 receptor, AS1411 and nucleolin, and SKN reduced the population of CSCs [132]. Another study showed that IGU loaded PLGA NPs (IGU-PLGA-NPs) were able to improve therapeutic outcomes in glioma, glioma stem-like cells, and temozolomide resistant glioma cells. It demonstrated not only significant inhibition of glioma cells proliferation both in vitro and in vivo but also anti-migration in glioma cells with low systemic toxicity [133].

Cell membrane-covered nanocarriers are also attractive delivery platforms. A recent study achieved DOX-Lex dual-drug delivery via angiopep-2 functionalized red blood cell membranes camouflaged NPs. The nanosystem showed prolonged blood circulation, superior BBB penetration, improved accumulations at the glioma site, and reduced tumor growth [130].

### 4.5. Hepatocellular Carcinoma (HCC)

HCC, the most frequent type of liver cancer, is a typical hyper-vascular tumor and shows a poor response to current conventional drug treatments [180]. In addition, more than ninety percent of patients relapse after treatment, which mainly accounts for the morbidity.

Surface molecules that are only highly expressed on the surface of hepatocytes and hepatocellular carcinoma cells, such as ASGP-R and glycyrrhetinic acid (GA) receptor, can be used as targets for nanodrugs delivery to increase cellular uptake. Thus, galactose residues were frequently used to modify the surfaces of NPs for selective hepatic delivery, due to their binding affinity to ASGP-R. Currently, the research of HCC targeting DDS is primarily focused on single ligand-modified polymeric NPs. For example, galactosylated chitosan NPs loaded with TP (GC-TP-NPs) were prepared and assessed in vitro and in vivo. GC-TP-NPs cellular uptake was greater than free TP in vitro and accumulated preferentially in the liver tumor in vivo. GC-TP-NPs were taken up by SMMC-7721 cells more than the non-modified NPs, indicating ASGP-R mediated endocytosis accelerated its uptake. Moreover, the TP is released from GC-TP-NPs in a sustained manner, which may contribute to maintaining higher TP concentrations for long periods at the tumor sites [135]. Dual or multiple coating of nanocarriers for increased efficiency and selectivity of the targeting delivery is also an interesting therapeutic trend. Accordingly, a dual-ligand system, 18β-GA and lactobionic acid (LA)-modified chitosan NPs loaded with DOX, displayed enhanced intracellular uptake of the drug. In vivo and in vitro studies also proved that the developed system showed an enhanced safety profile [141].

Besides, there have been also several attempts to treat HCC by co-loading dual drugs into a single nanocarrier. Dual DDS is being investigated to deliver Sf, which is the only available systemic drug for HCC. The combined therapy of pH-sensitive carboxymethyl chitosan-coated liposomes for delivery of Sf and siRNA against vascular endothelial growth factor (VEGF-siRNA) was found to significantly enhanced VEGF downregulating effect, inducing cell early apoptosis, as compared to free siRNA and single loaded carrier [136].

Researchers also investigated simultaneously encapsulating molecular inhibitors and standard cytotoxic chemotherapeutics within a single nanocarrier. Apa and pSN38 co-loaded NPs were formulated using mPEG5k-PLA8k, the clinically approved amphiphilic copolymer, for sequential delivery of both encapsulated drugs to HCC. Results indicated that stable NPs realized the sequential release of both encapsulated drugs to exert antimetastatic, antivascular, and cytotoxic activities simultaneously and reduced drug resistance [137]. CD44-targeted HA-conjugated Janus nanocarrier (HA-MSN@DB) for delivery of DOX and BER was found to significantly enhanced the antitumor activity of DOX and suppressed DOX-exacerbated HCC repopulation in vitro and in vivo. In addition, it was reported that the nanomedicine exhibited better intracellular internalization and favorable tumor accumulation [138]. In another study, the researchers used PEG-graft-polyglutamic acid (PEG-PLG) and HA-decorated NPs to simultaneously encapsulate DOX and diethyldithiocarbamate–copper complex (Cu(DDC)_2_) to achieve the selective co-delivery of the drug to HCC cells. It is worth noting that Cu(DDC)_2_ was not only an effective component for cancer therapy but also the composition of carrier materials. The NPs significantly improved the delivery of drugs to HCC cells, resulting in greater cellular uptake in HepG2 cells. At the same time, the results indicated the targeted NPs showed good synergistic effect and inhibited tumor growth in vivo [139]. In a recent study, lipids extracted from egg yolks have been used to prepare nano-sized particles. Application of folic acid (FA)-modified natural egg yolk lipid nanovector (EYLNs) load with DOX (FA-EYLNs-DOX) showed higher encapsulation efficiency and were effectively taken up by cancer cells without obvious toxicity in vitro and in vivo. In addition, FA-EYLNs-DOX showed that FA significantly achieved better tumor targeting of EYLNs-DOX [140].

Although many nanomaterials are currently in preclinical development, the number of nanomaterials in clinical trials and approved for clinical use is still low. Nanoformulations developed for selected solid tumors under clinical trials are summarized in Table 3. The formulations are mostly based on liposomes, polymeric NPs, and NP albumin-bound PTX (Abraxane^®^). In preclinical practice, nanodrugs generally increase tumor growth inhibition and prolong survival compared with non-formulated drugs, but in clinical settings, patients tend to benefit from nanomedicines due to reduced or altered side effects. Most clinically approved cancer nanomedicines are based on standard cytostatics, such as DOX, daunorubicin, PTX, VCR, and irinotecan [181]. Liposomal irinotecan (Onivyde^®^) has recently been approved in combination with 5-FU and leucovorin for the treatment of PC [182]. Abraxane^®^ combined with the immune checkpoint inhibitor atezolizumab, another promising systemic combination therapy, together induced unprecedented therapeutic responses in patients with triple negative BC [183]. Inspired by these results, recent clinical trials have tried to incorporate nanomedicines in systemic combination therapies. Suffice it to say, these preclinical studies manifest that there is no lack of innovative ideas and platforms for targeted drug delivery.

## 5. Conclusions

With the development of nanocarriers, the versatility of NPs enables them to promote drug synergy under precise control of drug distribution in space and time with the aim of reducing the risk of drug resistance and targeting different types of cancer. Many biocompatible nanometer material carriers have been studied and conjugated with multiple active pharmaceutical ingredients. However, with the increased complexity of the nanoformulation, it probably leads to higher toxicity, increased manufacturing cost, and manufacturing practice issues. More advanced technologies are needed to assess the interactions between nanomedicines and biological systems [184]. In addition, the loading concentration and encapsulation efficiency of active drugs, the two basic parameters used to evaluate nanocarriers, may be also the main disadvantages of some NPs delivery platforms. For instance, liposomes have a low encapsulate rate and a short releasing time [185]. More importantly, the rational combination of multiple drugs with different modes-of-action is also a challenge, since most DDS simply mixes different drugs together to co-administration, regardless of pharmacokinetics and distribution in tumor sites. In addition, when the NPs come into contact with live cells in vivo, proteins immediately covers the surface of the nanomaterials, affecting cell uptake, inflammation, accumulation, and degradation of NPs [186]. A great many nanocarriers have been developed, which establishes the foundation for its clinical translation. But while nanocarriers have many advantages, only a few of them have been approved by the FDA [187]. The accurate identification of patients suitable for clinical trial research, an in-depth understanding of their mechanisms of action, and the establishment of effective academic-industrial partnerships at all stages of drug development is important for the successful transition of new nanocarriers from pre-clinical to clinical. There are also several obvious challenges on the commercial and industrial scale, including reproducibility, non-uniform size, irregular structure/shape, sterilization, and storage for mass production [188]. The accumulation of nanodrugs in unwanted tissues and organs results in long-term toxicity problems. Therefore, in preclinical and clinical studies, the determination of the biological distribution of NPs after systemic administration should be considered.

Although there are still uncertain safety concerns of these nanomaterials, the current findings strongly suggested that nanomaterials have a promising future in the field of solid tumor treatment. In order to render nanocarriers preferably adaptable in vivo to the complex environment, more reasonable strategies for combining chemotherapy drugs, nanocarriers, and targeting moieties need to be developed.

## Figures and Tables

**Figure 1 cancers-12-02783-f001:**
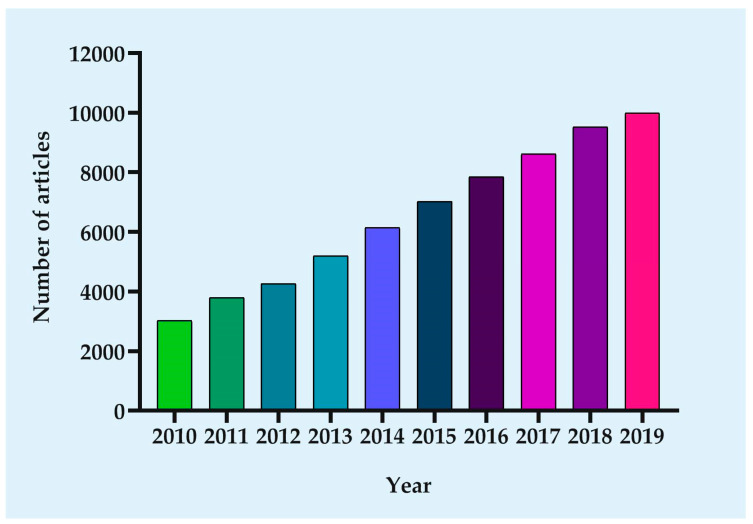
Growth in the number of published articles. Since the focus of this paper is on nanocarrier-based drug delivery for cancer therapy, the following parameters were searched in PubMed: ((cancer) OR (tumor)) AND ((((((((((((((nanoparticle) OR (nanocarrier)) OR (liposome)) OR (micelles)) OR (dendrimers)) OR (Niosomes)) OR (nanoemulsions)) OR (nanocrystals)) OR (Exosomes)) OR (quantum dots)) OR (carbon nanotubes)) OR (nanodiamonds)) OR (nanocapsules)) OR (hybrid nanomedicine)).

**Figure 2 cancers-12-02783-f002:**
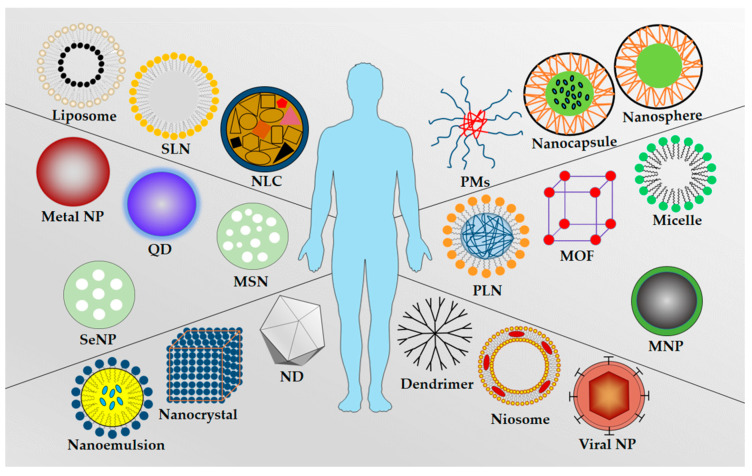
Schematic representation of the commonly used nanocarrier types: Liposome, solid lipid nanoparticle (SLN), nanostructured lipid carrier (NLC), polymeric micelles (PMs), nanocapsule, nanosphere, metal nanoparticle, quantum dot (QD), mesoporous silica nanoparticle (MSN), polymer–lipid hybrid nanoparticle (PLN), metal–organic framework (MOF), micelle, selenium nanoparticle (SeNP), magnetic nanoparticle (MNP), nanoemulsion, nanocrystal, nanodiamond (ND), dendrimer, niosome, and viral nanoparticle.

**Figure 3 cancers-12-02783-f003:**
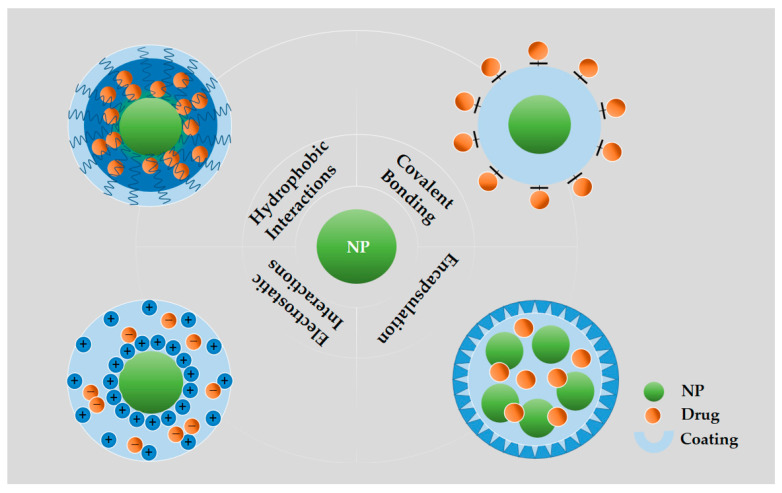
Illustration of various methods of loading/bonding therapeutics into NPs. Covalent bonding: Drugs directly to the surfaces of NPs through covalent bonds; Hydrophobic interactions: Partitioning hydrophobic drug molecules in an amphiphilic corona layer; Electrostatic interactions: Loading drugs onto the surfaces of NPs by electrostatic layer-by-layer assembly; Encapsulation: Loading drugs into the hollow NPs.

**Figure 4 cancers-12-02783-f004:**
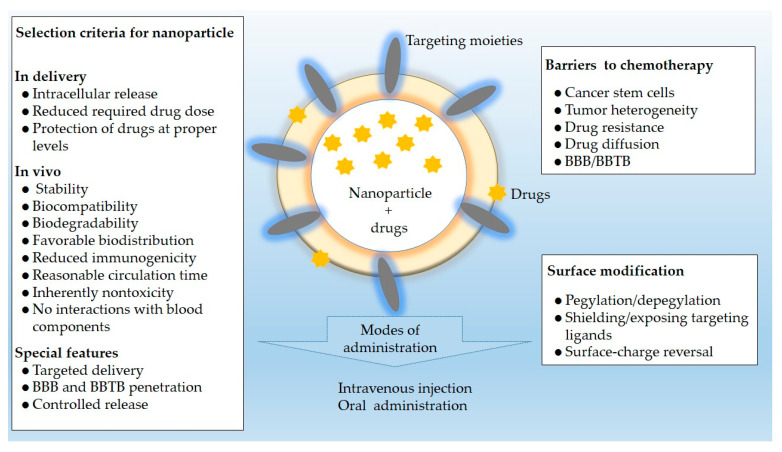
Selection criteria, barriers to chemotherapy, and surface modification associated with NPs for drug delivery in selected solid tumors mentioned in this review.

**Table 1 cancers-12-02783-t001:** Summary of types of NPs and their advantages and limitations.

Nanosystems	Advantages	Limitations
**Liposomes**	Biodegradability Biocompatibility Reduced systemic toxicity Improved stability and circulation time of the drugs	Low drug loading Lack of colloidal stability Difficulties in sterilization Some leakage of the encapsulated agent
**SLNs**	Suitable for a variety of routes of administration Good physiological compatibility A wide range of drug adaptability Improve the stability of drugs	Requires organic solvent during preparation Low loading capacity compared with other nanocarriers Possibly containing other colloidal structures and complex physical state
**NLCs**	Stability Good biocompatibility High drug-loading capacity Targeting and controlled release Improved bioavailability of drugs	No data
**Micelles**	Suitable for water-insoluble drugs due to hydrophobic core	Poor chemical versatility and structural instability
**PMs**	Good stability Allow drugs to avoid mononuclear macrophages phagocytosis Water-insoluble drugs can be easily incorporated into PMs by the simple act of mixing	Premature drug leakage Toxicity of materials, fixed functionality after synthesis Response mechanism in the human body unknown
**Polymeric NPs**	Targeted, controlled drug release Easy surface functionalization	The polymer cytotoxicity Difficulty in large-scale industrial production The residual organic solvent in the preparation process
**Dendrimers**	Structural symmetry and stable nature Has a strong EPR Enhancement in the blood circulation time Multiple functional groups in its surface Customize the drug release profiles	Time-consuming synthesis and increased production costs Difficulties in mass production Cytotoxic and hemolytic properties
**Niosomes**	Overcome phospholipid oxidation	No data
**Nanoemulsions**	Improve drug stability To avoid drug inactivation in the gastrointestinal tract Increase drug solubility and improve bioavailability With lymphatic targeting and sustained release to reduce the side effects of drugs	No data
**Nanocrystals**	Free of organic solvents or other solubilizing chemicals Carrier-free delivery system High-drug loading efficiency Steady dissolution rates Great structural stability	Difficulty in large-scale industrial production
**Bio-NPs**	Overcome various biological barriers Lower immunogenicity and toxicity Biocompatibility and biodegradability	Limited drug loading capacity
**Exosomes**	High capacity to pass through various biological barriers High stability Less immunogenity Natural targeting capacity	Specific approaches to load the desired additional drugs without disturbing the exosomes
**Metal NPs**	Simple synthesis procedures Modifiable (control of pore size) Multifunctional surface functionalization	Poor biocompatibility Low stability Poor water solubility
**Inorganic non-metallic NPs**	Simple synthesis procedures	Low loading capacities
**Hybridluhan** **nanomedicines**	Targeted delivery of drugs Has high structural integrity Stable storage of drugs and the controlled release of drugs Increased drug encapsulation efficiency and biocompatibility	Potential material toxicity

**Table 2 cancers-12-02783-t002:** A brief summary of the application of nano-based DDS in selected cancers.

Studying Group [Ref.]	NPs Description	Targeting Mechanism	Target	Drug(s)	Cancer Model	Results of Findings
Kim et al. (2016) [96]	Exosome	Specific endocytosis and/or fusion with plasma membranes	No data	PTX	3LL-M27 cells; MDCK_WT_ and resistant MDCK_MDR1_ cancer cells; pulmonary metastases in Lewis lung carcinoma (LLC) mouse model	The incorporation of PTX into exosomes significantly increased drug accumulation levels in both sensitive and resistant cancer cells; a significant (*p* < 0.05) inhibition of metastases growth by exoPTX treatment was demonstrated
Iannazzo et al. (2017) [97]	QDs	Active targeting	Biotin receptors	DOX	A549 cells	Improved delivery of conventional chemotherapeutics by using QDs as nanocarrier
Zhao et al. (2017) [98]	Micelles	Passive targeting	EPR	PTX and DOX	A549 cells	A fixed and high drug loading content of 24.2% (PTX~14.8% and DOX~9.4%) with a precise ratio of PTX and DOX to realize the synchronized and controlled release
Xie et al. (2019) [99]	PMs	Not mentioned	No data	PTX and ligustrazine	A549 cell lines; xenograft tumor mice model	Strong inhibition on tumor metastasis; enhance the accumulation of drugs at tumor sites; tumor volume ratios were 26.47% ± 8.23 for blank control, 21.43% ± 9.45 for free PTX, 14.65% ± 8.13 for dequalinium (DQA) modified PTX plus ligustrazine micelles, respectively
Zou et al. (2019) [100]	SeNPs	Specific endocytosis	CD44 receptor	PTX	A549 cell lines; A549 tumor-bearing mice	Greater uptake of PTX in A549 cells; negligible toxicity; PTX and HA-Se@PTX at 4 μg/mL PTX dose dramatically inhibited the proliferation of A549 cells and the cell viability rates were 64.8, and 34.5%, respectively
Wu et al. (2019) [101]	Dendrimers	Coating with targeting cancer cell membrane proteins	No data	DOX and icotinib	H1975, HCC827, and B16 cell lines; H1975 tumor-bearing mice	High stability; superior targeting ability; minimal side effects; at the physiological pH 7.4, only 30.1% of the DOX and 27.3% of icotinib were released from the dendrimers within 48 h; the H1975 cell membrane-coated dendrimers resulted in 87.56% tumor inhibition, with the tumor weight 8.75-fold less compared to that of the PBS control group
Huang et al. (2019) [102]	PMs	Active targeting	CD133 and CD44 receptor	Gefitinib	H446 and A549 cell lines; xenograft tumor mice model	The drug loading of the nanomicelles in each group was 7–9% and the encapsulation efficiency was ~80%; exhibited greater therapeutic efficacy against lung cancer-initiating cells than single-target
Zhang et al. (2019) [103]	Polymeric NPs	Specific endocytosis	Epidermal growth factor receptor (EGFR)	Homoharringtonine (HHT)	BEAS-2B, A549, and NCI-H226 cell lines; A549 tumor bearing mice	Better therapeutic efficacy and fewer side effects; targeted recognition and stimuli response; the IC50 of the nanomedicine is 5.1 nM, while the IC50 of free HHT reaches up to 23.2 nM, a 4.5-fold increase
He et al. (2020) [104]	PMs	Not mentioned	No data	PTX	A549 cells; A549 tumor bearing mice	Enhanced the retention of drugs in the tumor; sustained drug release property; the IC50 values of the PTX micelles at 24 h with no ambroxol (Ax) or combined with 100 μM Ax were 87.09 ± 4.12 ng/mL and 1.14 ± 0.08 ng/mL, respectively
Liang et al. (2020) [105]	NLCs	Specific endocytosis	Glucose	PTX and GEM	LTEP-a-2, L929, and A549 cell lines; A549 tumor bearing mice	Targeted intracellular sequential drug release; the tumor volume in dual-drugs-loaded NLCs group was 2.6-fold smaller than those treated with the free drug combination
Zhang et al. (2015) [106]	Nanocrystals	Not mentioned	No data	PTX	MDA-MB-231/Luc cells; MDA-MB-231/Luc tumor bearing mice	PEGylated PTX nanocrystals significantly enhanced the antitumor effect in treating in situ tumor or metastatic tumor bearing mice after intravenous administration
Guven et al. (2017) [107]	CNTs	Passive targeting	EPR	Cisplatin	MCF-7 and MDA-MB-231 tumor bearing mice	A prolonged circulation time compared to free cisplatin which EPR effects resulting in significantly more cisplatin accumulation in tumors
Li et al. (2017) [108]	PLNs	Active targeting	Human epidermal growth factor receptor-2 (HER-2)	Salinomycin	BT-474 ALDH+ and ALDH- cell; MDA-MB-361 ALDH+ and ALDH- cells; BT-474 tumor bearing mice	Achieved the best therapeutic efficacy, resulting in a 79% decrease in tumor volume, whereas salinomycin obtained only moderate therapeutic efficacy (43% decrease)
Le et al. (2017) [109]	Viral NPs	Not mentioned	No data	DOX	MDA-MB-231 cells; MDA-MB-231 tumor bearing mice	DOX-loaded viral NPs demonstrated efficacy in MDA-MB-231 cell although at lower efficacy than free DOX
Jiang et al. (2018) [110]	SiNPs	Not mentioned	No data	DOX	EMT-6 and MCF -7 cell lines; EMT-6 tumor bearing mice	The tumor size and weight of DOX loaded SiNPs group was 2-fold and 1.7-fold smaller than that of free DOX group, and 4-fold and 2-fold smaller than that of PBS group
Zheng et al. (2019) [111]	SLNs	pH sensitivity	No data	DOX	MCF cells lines; MCF/ADR DOX-resistant cells; MCF/ADR tumor bearing mice	RGD-DOX-SLNs showed 5.58 fold higher area under the plasma concentration-time curve (AUC) compared with DOX solution; terminal half life (T_1/2_) and peak concentration (C_max_) of RGD-DOXSLNs was 10.85 h and 39.12 ± 2.71 L/kg/h
Fang et al. (2019) [112]	Polymeric NPs	Active targeting	CD44 receptor	DTX	4T1-Luc cells lines; 4T1-Luc tumor bearing mice	Drug loading efficiency (76.3−80.4%); steady in a nonreducing environment while was destabilized under 10 mM glutathione releasing ~90% of the DTX within 24 h; selective cellular uptake
Li et al. (2019) [113]	Liposome	Passive targeting	EPR	Poria cocos extract and DOX	MCF cells lines; MCF/ADR DOX-resistant cells; MCF/ADR tumor bearing mice	Higher safety; sensitized DOX to kill cells in drug-resistant tumors; the release rates of poria cocos extract from the liposome were > 70% within 6–8 h, while DOX was released completely after 12 h
Lei et al. (2019) [114]	MOFs	Passive targeting	EPR	DOX	4T1, MDA-MB-231, MCF-7, and ZR-75-30 cell lines; 4T1 tumor bearing mice	Good safety profile; highly effective antitumor ability
Dancy et al. (2020) [115]	Polymeric NPs	Active targeting	Fibroblast growth factor–inducible 14 (Fn14) receptor	PTX	231-Luc cell lines; 231-Luc tumor-bearing mice; 231-Br-Luc tumor-bearing mice	Tumor cell–specific uptake; long blood circulation time; excellent tumor tissue penetration; the average tumor doubling time in the NPs treated mice was 32 days compared to 17 and 20 days for saline- or Abraxane-treated mice, respectively
Han et al. (2020) [116]	Liposomes	Specific endocytosis	ERs	PTX	MCF-7 cell lines; MCF-7 tumor bearing mice	Encapsulation efficiency of 88.07 ± 1.25%; prolonged half-life of the drug; the elimination half-lives of PTX and PTX liposomes were 1.79 and 20.98 h, respectively
Zafar et al. (2020) [117]	LNCs	Passive targeting	EPR	DTX and THQ	MCF-7 and MDA-MB-231 cell lines; Ehrlich ascites carcinoma bearing mice	Encapsulation efficiency of DTX and THQ were found to be 86.79 ± 3.79% and 95.17 ± 1.61%, respectively; controlled drug release; re-sensitized cancer cells to DTX; a 2.85-folds decrease in tumor volume was observed with LNCs treated group compared to control group
Xu et al. (2020) [118]	PMs	Active targeting	Sialic acid residues	DOX	MCF-7/ADR cell lines; MCF-7/ADR tumor bearing mice	MDR reversal; good stability in neutral environment; ~50% MCF-7/ADR cells were killed with DOX micelles treated compared to ~15% cells death induced by free DOX
Guo et al. (2020) [91]	RNA NPs	Active targeting	EGFR	PTX	MDA-MB-231 cell lines; MDA-MB-231 tumor bearing mice	Undetectable toxicity or immune stimulation; the in vitro cell apoptosis assay revealed that 45.1% of the cells underwent apoptosis after 24 h treatment with RNA NPs, in comparison to free PTX (24.6%)
Teijeiro-Valiño et al. (2018) [119]	Polymeric NPs	Active targeting	CD44 receptor	DTX	A549 lung cancer cells; orthotopic lung cancer model; PC patient derived xenograft model	Dual targeting properties (to the tumor and to the lymphatics); a dramatic accumulation of DTX in the tumor (37-fold the one achieved with Taxotere^®^)
Lin et al. (2019) [120]	Liposome	Specific endocytosis	EGFR	GEM and HIF1α-siRNA	PANC-1 cell lines; PANC-1 tumor bearing mice	Increased targeting specificity of liposome carrier; increased the total amount of apoptosis cells; GE-GML/siRNA showed 4-fold reduction in tumor compared to control group
Madamsetty et al. (2019) [121]	NDs	Passive targeting	EPR	DOX	BxPC3, 6741 and PANC-1 cell lines; orthotopic PDAC xenograft model	A considerable improvement over free drug; no abnormalities of major organs; NDs alone showed no cytotoxicity at doses up to 25 μg/mL, irrespective of whether the cells were grown in the absence or presence of FBS
Massey et al. (2019) [122]	Polymeric NPs	Not mentioned	No data	PTX	AsPC1, PANC-1, MIA PaCa-2, and HPAF-II cell lines	NPs administration (10 mg/kg) significantly (P << 0.05) inhibited tumor growth, even in pre-exposed mice as determined by significant (P << 0.05) inhibition of bioluminescence counts ideal properties for nano-scale drug delivery;
Madamsetty et al. (2019) [123]	NDs	Passive targeting	EPR	Irinotecan and curcumin	AsPC-1 and PANC-1 cells; orthotopic PDAC xenograft model	Exerted immunomodulatory effects; dual payload
Sun et al. (2020) [124]	PMs	Passive targeting	EPR	NLG919 and PTX	PANC02 and H7 cell lines; PANC02 tumor bearing mice; 4T1 BC model	Improved tumor inhibition effect; more immunoactive tumor microenvironment; micelles showed a more favorable release kinetics of PTX, and only 35% of PTX was slowly released within 72 h
Etman et al. (2020) [125]	Polymeric NPs	Specific endocytosis	Lactoferrin (Lf) and CD44 receptors	Quinacrine (QC)	PANC-1 cell lines; orthotopic PC model	pH triggered release; the loading efficiency of the dual coated formulation was 19.5 ± 1.9% compared to 23.6 ± 2.4% for uncoated formulation.
Elechalawar et al. (2020) [126]	Au NPs	Active targeting	EGFR	GEM	PANC-1, AsPC-1, CAF-19, and HPDE cell lines	Enhanced cellular uptake and cytotoxicity to pancreatic cancer cells (PCCs)
Han et al. (2020) [127]	MNPs	Active targeting	No data	GEM	PANC-1 and HUVEC cell lines; PANC-1 tumor bearing mice	Targeted delivery and effective accumulation; the GEM-loaded MNPs exhibited higher cytotoxicity at pH 6.5 than that at pH 7.4, which might be attributed to pH-dependent enhanced cellular uptake
Zhai et al. (2018) [128]	APO	Specific endocytosis	Transferrin receptor 1 (TfR1) and heparan sulfate proteoglycan	Vincristine sulfate (VCR)	bEnd.3, HUVEC, and U87MG cell lines; U87MG tumor bearing mice	Higher glioma localization; the VCR encapsulation efficiency was approximately 39.8 ± 0.9%; treatment with this NPs significantly prolonged the median survival time (35 days), which was 1.8 and 1.6-fold higher than that of physiological saline and free VCR, respectively
Guo et al. (2018) [129]	PMs	Specific endocytosis	IL-13R	CMS	BT325 cell lines; Luc-BT325 tumor bearing mice	BBB penetration; targeting glioma cells; the apoptosis rate of BT325 cells induced by the PMs nearly 80%
Zou et al. (2018) [130]	Polymeric NPs	Specific endocytosis	Lipoprotein receptor related protein receptor	DOX and lexiscan (Lex)	U87MG tumor bearing mice	Improved blood circulation time; BBB penetration; the biodistribution of nanomedicines demonstrated that orthotopic brain tumor accumulation was 21.9 fold higher than that of free DOX controls
Meng et al. (2019) [131]	PMs	Not mentioned	No data	DOX	HBMEC and C6 cell lines; GBM-bearing mice model	The drug encapsulation efficiency and loading capacity in DOX BO-PMs were (95.69 ± 0.49)% and (14.62 ± 0.39)%, respectively; enhanced the transport efficiency of DOX across the BBB; exhibited a quick accumulation in the brain tissues
Wang et al. (2019) [132]	Nanoemulsion	Active targeting	CD44 and nucleolin	Shikonin (SKN) and DTX	U87 cell lines; orthotopic luciferase-transfected-U87 bearing nude mice	BBB penetration; overwhelming inhibition of the orthotopic luciferase-transfected-U87 glioma-bearing nude mice; after incubating cells for 8 h, the nanoemulsion induced apoptosis in 71.3 ± 4.2% of U87 cells
Younis et al. (2019) [133]	Polymeric NPs	Not mentioned	No data	Iguratimod (IGU)	U87, U118, and U251 cell lines; xenograft tumor mice model	Without any visible organ toxicity; significant inhibition of tumor growth; cross BBB
Caban-Toktas et al. (2020) [134]	Polymeric NPs	Not mentioned	No data	R-flurbiprofen and PTX	RG2 cell lines; Rat RG2 glioma tumor model	Reduced inflammation in the peri-tumoral area; enhanced anti-tumoral activity against glioma
Zhang et al. (2018) [135]	Polymeric NPs	Specific endocytosis	Asialoglycoprotein receptors (ASGP-R)	Triptolide (TP)	SMMC7721 and A549 cells; HCC xenograft mouse model; orthotopic HCC mice model	Sustained release; targeted delivery; high liver tumor accumulation in vivo
Yao et al. (2019) [136]	Liposome	Not mentioned	No data	Sorafenib (Sf) and VEGF-siRNA	HepG2 cells; H22 tumor-bearing mice	Improved anti-tumor efficiency
Han et al. (2019) [137]	Polymeric NPs	Not mentioned	No data	Polymeric SN38 prodrugs (pSN38) and apatinib (Apa)	Huh-7, LM3, and HepG2 cell lines; HCC xenograft mouse model	Reduced drug resistance; the sequential release of both encapsulated drugs
Zhang et al. (2019) [138]	MSNs	Specific endocytosis	CD44 receptor	DOX and berberine (BER)	HepG2, H22, HL-7702, HCC cells, and NIH-3T3 cell lines; H22 tumor-bearing mice	Efficient tumor-inhibiting effects; decreased regrowth activity; the apoptotic rates of DOX+BER and DOX+BER loaded MSNs were 34.93 and 48.10%, respectively
Xu et al. (2019) [139]	Oxide NPs	Specific endocytosis	CD44 receptor	DOX and Cu (DDC)_2_	MCF-7 and HepG2 cell lines; mouse models of ectopic hepatocellular carcinoma	Improved stability; specific targeting of HCC; good synergistic effect; the tumor volume and tumor weight of the oxide NPs treated group reduced to 60.32% and 60.39% compared to the control group, respectively
Tang et al. (2020) [140]	Liposomes	Active targeting	Folate receptor (FR)	DOX	4T1 cell lines; H22 and Eca9706 tumor-bearing mice	High drug load capacity; effectively taken up by cancer cells; no obvious toxicity
Hefnawy et al. (2020) [141]	Polymeric NPs	Active targeting	ASGP-R	DOX	Hep-G2 cell lines; HCC-bearing rats	Improved intracellular drug delivery and uptake; enhanced safety profile; the ability of the NPs system to enhance the intracellular uptake of the drug by 4-fold and 8-fold after 4 h and 24 h of incubation, respectively

**Table 3 cancers-12-02783-t003:** A list of recent clinical trials of nanomedicines for the treatment of selected solid tumors.

Particle Type/Therapeutic Agent	Treatments	Cancer Subtypes	Trial Starting Date	Phase	Aim of the Study	NCT Number
ABI-009 (nab-Rapamycin)	Combination therapy	GBM	Aug. 2018	II	ABI-009 will be tested as single agent or in combination with standard therapies	NCT03463265
Abraxane^®^	Combination therapy	pancreatic ductal adenocarcinoma (PDA)	Jul. 2018	II	To compare the first-line treatment with nab-PTX plus S-1 and nab-PTX plus GEM in advanced PDA with primary tumor nonexcision in Chinese patients	NCT03636308
Abraxane^®^	Combined with CPT	HER-2 Negative BC	Nov. 2018	IV	To evaluate of the efficacy and safety of nanoparticle albumin-bound PTX combined with CPT as neoadjuvant chemotherapy in luminal B/HER-2 negative BC	NCT03799692
Abraxane^®^	Combined with Epirubicin and Cyclophosphamide	TNBC	Nov. 2018	IV	To evaluate of the efficacy and safety of weekly Nab-P followed by dose-intensive epirubicin in combination with cyclophosphamide as neoadjuvant chemotherapy in TNBC	NCT03799679
Pegylated liposomal DOX (PLD)	Combined with trastuzumab	HER2-positive BC	Mar. 2019	II	To evaluate the efficacy and safety of PLD in combination with trastuzumab in HER-2 positive metastatic BC	NCT03933319
Pegylated Liposomal DOX (Doxil/Caelyx)	Combined with pembrolizumab (Keytruda)	Metastatic Endocrine-resistant BC (ERBC)	Apr. 2019	I/II	To evaluate the tumor response and appropriate dose of a chemo-immunotherapy regime consisting of treatment with PLD and pembrolizumab-based in ERBC patients	NCT03591276
QDs coated with veldoreotide	Monotherapy	BC	Sep. 2019	I	A novel formulation for treatment and bioimaging of BC which can deliver safely to the patients in a high dose to the affected tumor cells	NCT04138342
Pegylated liposomal DOX (PLD)	Combined with albumin-bound PTX and trastuzumab	HER-2 positive BC	Oct. 2019	I/II	To evaluate the efficacy and safety of PLD plus Albumin-Bound PTX and trastuzumab as neoadjuvant therapy in HER-2 positive BC	NCT03994107
Abraxane^®^	Combined with CPT	TNBC	Dec. 2019	III	This trial will compare the therapeutic effect of albumin-bound PTX with solvent-based PTX in TNBC patients, and seek for important scientific clues, scientific evidence, and clinical data for nab-P in the treatment of TNBC	NCT04137653
PTX liposome	Combined with S-1	Advanced PC	Jan. 2020	IV	To investigate the efficacy and safety of the patients with confirmed advanced PC after treating with the combination of PTX liposome plus S-1	NCT04217096
Abraxane^®^	Combined with Alpelisib	TNBC	Feb. 2020	II	To determine if alpelisib in combination with nab-PTX will improve treatment effect of patients with chemotherapy insensitive TNBC	NCT04216472
Liposomal irinotecan (nal-IRI)	Combined with Oxaliplatin, Leucovorin, and 5-Fu	Locally Advanced Pancreatic Carcinoma (LAPC)	Mar. 2020	II	To investigate the efficacy and tolerability of a combination of liposomal irinotecan (nal-IRI) with oxaliplatin, leucovorin, and 5-Fu (FOLFOX-nal-IRI) for treatment of patients with LAPC	NCT03861702
Liposome-entrapped mitoxantrone hydrochloride injection (PLM60)	Monotherapy	Advanced HCC	Apr. 2020	I	To evaluate the safety and efficacy of PLM60 in advanced HCC	NCT04331743
NanoPac (sterile nanoparticulate PTX) powder for suspension	Monotherapy	SCLC	May 2020	II	To evaluate the use of NanoPac injected directly into tumors in the lung of people with lung cancer	NCT04314895

Data were gathered by searching the National Institutes of Health (NIH)’s Clinical Trials.gov database at https://clinicaltrials.gov/. This table includes information on clinical trials as of 19 May 2020.

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
