# Peer review of "Recent Progress of Nanocarrier-Based Therapy for Solid Malignancies"

_cancers, 2020, doi:10.3390/cancers12102783_

Round 1

Reviewer 1 Report

The manuscript authored by Wei, Q. et al entitled " Recent Progress
of Nano carrier Based Drug Therapy for Solid Malignancies " has reported an overview of the recent development of nanocarriers, and the updated status of their use in the treatment of several solid tumors. The authors in this manuscript discuss the using these nanocarriers and their utilization for in vitro and in vivo studies. Overall authors did an excellent job in addressing most of my previous comments. 

However, this work still needs minor revision and responding to some concerns as below:

1- Number of figures are still very low. Could the authors add more figures from the good research papers they represented in their references. Taking copyright permission or drawing could be a possible option.

2- Paper should be revised for some nanotechnology terminology such as such as Section 2.8. Bio-NPs?? Polymeric micelles??

3- Tables should include examples for at least all NPs. listed in the review. 

4- Polymeric micelles! I think polymeric NPs are different than micellar formulations. So it should be modified or separated to different sections.  

Reviewer 2 Report

The review of Wei et al. is devoted to the emerging topic of anti-tumor therapy, namely to the nano medicine approaches to the treatment of solid malignancies. At the previous stage, there were several points to improve. The resubmitted version is of much better quality, the authors have done a nice work to organize the material properly, to improve the readability and comprehension of material. However, in this version, there are some moments to pay attention form. My comments are given below.

  1. In the title, the authors claim to review "recent progress", that commonly means 5-10 years span. Considering the intensive development of the field (that is mentioned in the Fig.1), I suggest to restrict the scope to the time span 2015-2020 and to revise the material and references accordingly. 
  2. The phrase "nanocarrier-based drug therapy" makes no sense, it is more commonly known as "Nanomedicine". As an alternative, "nanocarrier-based therapy" could be a solution. The authors are advised to address to the literature on the topic for better terms.
  3. In the Section 2, being, in fact, an introduction for the readers to the variety of nanoconstructions, I recommend to cite mainly reviews, not  experimental papers.
  4. In the Section 4, clinical trials have been mentioned in the title, however, the table summarizing their findings has disappeared from the revised version. I think it is worth being retuned to the text and properly discussed.
  5. In the Table 2, I suggest to use "results" of "findings" instead of "advantages"
  6. The text needs proofreading for correct word usage

Reviewer 3 Report

Some comments referred in the first revision have been ammended, although I do not sense a strong background of authors on nanomedicine.

You have improved the information regarding nanomedicine, but it is clear you are not experts in the field, since references included do not represent the experts on each field. 

For example: Couvreur, the first one that injected nanoparticles in animals, is only referred in a non-key reference; Solans and Fornaguera, the group that has published more on nano-emulsions, which, by the way is not well written in your manuscript, are not even referred. And what about Langer, Kabanov, Kreuter.... they are key in the field and not mentioned.

Round 2

Reviewer 2 Report

The authors have done a nice work to improve the quality of the manuscript. I think it can be published after language polishing and proofreading. I have no further comments for this submission. 

Reviewer 3 Report

No more comments

This manuscript is a resubmission of an earlier submission. The following is a list of the peer review reports and author responses from that submission.

Round 1

Reviewer 1 Report

The manuscript authored by Wei, Q. et al entitled " Recent Progress of Nano Based Drug Delivery System for Solid Malignancies " has reported an overview of the recent development of nanocarriers, and the updated status of their use in the treatment of several solid tumors. The authors in this manuscript discuss the using these nanocarriers and their utilization for in vitro and in vivo studies. Overall authors did an excellent job in reviewing the field critically and presented the most recent studies about all subtopics. However, I find some flaws in the manuscript that could be addressed. Because the motivation is clear, and the authors have conducted a well-documented review in various tumors. I suggested accepting this work with major revision and responding to some concerns as below:

1-The manuscript has several punctuation and grammatical errors.  It has to be proofread or checked again. 

2- The title did not represent the review content. Since the review only discuss chemotherapeutic agents and some drug delivery systems (no mentions of biological drugs, gene therapy, or vaccines) , could they modify it to represent the real contents?

3- Page 5, section 2.5. Nanoemulsions: these sentences have no references. References are required.

4-There is some redundancy like page 6 “Protein-based nanocarriers have several advantages in drug delivery, such as biodegradability, good biodegradability, easy surface modification ability, no immunological responses, and low toxicity.”.

5- Many sentences in exosomes part in page 6 were not clear such as “It is also important to use a particular technology to load a variety of additional drugs and to target skills without disturbing the exosomes, as damaged exosomes with no exosomal signals may lead to undesirable immune responses. Understanding the function and structure of exosomes will contribute to the clinical setting of exosome therapy [50].” Also, can you add more about what is the current clinical setting for exosomal delivery of chemotherapy?

6- In page 9, can you clarify this statement” Recently, a study was conducted using PTX and gemcitabine (GEM)-conjugated NLCs on the NSCLC cell lines to determine its efficacy against cancer. In vitro release studies showed that first PTX (redox-triggered), then GEM (pH-triggered). In addition, this study indicated this nanomedicine achieved a synergistic antitumor effect by targeted intracellularly sequential drug release in vivo [72].”

7- Figure 1. Font need to be unified.

8-Number of figures are very low. Could the authors add more figures from the good research papers they represented in their references. Taking copyright permission or drawing could be a possible option.

9- could the authors add a small separate section about current clinical trials table progress? They can show what are the advancements and how these NPs selection (mentioned in the table 2.) improved the current clinical settings of these medications and progressed to further clinical trials phases. 

Reviewer 2 Report

The review of Wei et al. is devoted to the emerging topic of anti-tumor therapy, namely to the nano medicine approaches to the treatment of solid malignancies. The review brings together chemical overview of nano carriers used for the anti-tumor drugs, biological features of target tumors and clinical aspects (clinical trials). The text is well-written and concise, nevertheless, certain proofreading is recommended. I believe this review to be interesting for readers of Cancers, however, I recommend to elaborate on certain points:

  1. I do not recommend to use the term "nano-based" in the title. "nanoscale" or "nano-sized" would be more precise. "Drug delivery systems" are better plural.
  2. The authors may wish to elaborate better on chemical aspects of drug delivery systems. How drugs are bound to the nanocarriers? It can be conjugation or complexation or encapsulation, each kind of loading has advantages and disadvantages. However, in the text, these features are not properly addressed. This information would strengthen the Section 2 and Table 1. 
  3. In the Section 2, choosing the nanoparticles to review should be justified: there are many types of nanoparticles used as drug carriers for tumor treatment, some of them have not been included in the review. 
  4. It would be nice to clearly describe advantages and disadvantages of each kind of nanocarriers described. 
  5. It seems quite logical that nanocarriers listed in the Section 2 would appear in the Table 1. However, not all species appear in the Table.
  6. Some subsections in the Section 2 contain too limited information (for example, 2.4-2.6). For example, section 2.4 contains just two references, though there are numerous recent papers and reviews on the use of dendrimers as drug carries for tumor treatment.
  7. I recommend to elaborate on the tumor-specific nanoparticle targeting. Are there tumor-specific ligands? How they can be/are used in the design of anti-tumor nanoconstructions? How efficient they are? This information would be suitable for the Section 3, if divided by the tumor types, and quite useful for readers. 
  8. The claimed topic of the review is the treatment of solid tumors. However, in the Table 1, the authors give some examples of the use of nanodrugs just in vitro, without further validation in vivo: Refs. 60, 64, 71, 75, 77, 85, 102, 106, 109. What is the purpose of including those refs into the table? 
  9. It seems that a separate Discussion section is needed. It would be useful to discuss there above-mentioned advantages and drawbacks of the drug delivery systems as well as some unsolved problems, perspectives in commercialization and clinical testing etc. 

The manuscript submitted does not contain line numbering that makes addressing to a given phrase difficult. 

Reviewer 3 Report

Wei et al has presented a comprehensive literature review on the nanotherapy based drug delivery systems in cancers. 

This review article collectively describes various nano based system and their applications in solid cancer therapeutics.

The article is well written with adequate use of references.

This reviewer, therefore, does not have any major concerns or comments on the quality of the review article, however, have couple of suggestions if authors could implement to emphasize the significance of the article.

  • A chart or graph should be included showing how many peer reviewed articles have been published on nano based drug delivery in cancers, years wise or in any format. This would highlight the importance of this topic in the field.
  • A thorough proof read is needed to eliminate any grammatical and typographical errors.

Reviewer 4 Report

This review deals with an important issue: the use of nanosystems for therapeutics in solid tumors, which I found of great interest. For the readers of Cancers journal. However before publication, I recommend the following modifications, since in the current form it does not fulfill the requirements of Cancers journal:

  • The title must be more accurate, since in the text you only talk about chemotherapeutic drugs, this must be understood with the title.
  • In general, the references are very poor, and the content a little bit too…. There are many authors that work with nano-chemotherapeutics and you did not refer them. In addition, you did not get deep into the important concepts that need to be highlighted in a chemotherapy review. Your articles seems too divulgater for a recognized journal as cancers is. Please, the following issues need to be amended before publication:
    • Which are the different chemotherapeutic drugs?
    • Which differential properties exist between different cancer types?
    • Only chemotherapy is used for solid tumors treatment?
    • Active targeting strategies need to be better detailed.
  • PEG is introduced without describing it. In addition, you introduced it into liposomal formulations while it is a polymer. This is an error. In addition, are you sure that PEGylation is appropriate right now (I agree that with the knowledge of some years ago yes) for cancer therapeutics?
  • You state the use of solvents as a drawback for SLNP, but they are also used for the preparation of liposomes, for example,
  • Niosomes is a very tight group. Do you really think that it is important to be highlighted in the same level than liposomes or polymeric nanoparticles? For sure the answer is not!
  • This is not true: Polymeric nanosystems are a more stable structure than liposomal systems in the form of nanospheres (solid spheres) or nanocapsules (hollow spheres with a void space in the center), where chemotherapeutic agents are encapsulated or conjugated onto the surface by covalent or non-covalent means
  • A polymeric nanoparticles CANNOT be composed of proteins, because in this case, it is not a polymeric nanopartivle.
  • Micelles are a group by themselves.
  • No references of nano-emulsions? You invented the term?
  • Bio-NP is an invented term, not common in nanomedicine.

I stop reading. Your knowledge on nanosystems is really poor.

Reviewer 5 Report

Nanotechnology for drug delivery has garnered tremendous and consistent attention in the oncology world because of its potential to deliver drugs accurately and precisely, with a potentially longer shelf life and reduced adverse effects. Therefore this review is timely and relevant.

However, the study suffers from multiple major drawbacks such as

  1. copious lack of proper citations
  2. use of qualitative instead of quantitative vocabulary to explain the superiority of nano-formulations over non-nano-based therapies

I tried to point out as many as of these anomalies as possible but this draft needs to revised and re-written to be made sense of. I suggest a full re-write before it is re-submitted for review. Please find my detailed review attached.
